# RIM: Reliable Influence-based Active Learning on Graphs

**Wentao Zhang**[1,2]**, Yexin Wang**[1]**, Zhenbang You**[1]**, Meng Cao**[2]**, Ping Huang**[2]
**Jiulong Shan**[2]**, Zhi Yang**[1,3]**, Bin Cui**[1,3,4]
[1]School of CS, Peking University [2]Apple
[3] National Engineering Laboratory for Big Data Analysis and Applications
[4]Institute of Computational Social Science, Peking University (Qingdao), China
[1]{wentao.zhang, yexinwang, zhenbangyou, liyang.cs, yangzhi, bin.cui}@pku.edu.cn
[2]{mengcao, Huang_ping, jlshan}@apple.com

## Abstract

Message passing is the core of most graph models such as Graph Convolutional Network (GCN) and Label Propagation (LP), which usually require a large number of clean labeled data to smooth out the neighborhood over the graph. However, the labeling process can be tedious, costly, and error-prone in practice. In this paper, we propose to unify active learning (AL) and message passing towards minimizing labeling costs, e.g., making use of few and unreliable labels that can be obtained cheaply. We make two contributions towards that end. First, we open up a perspective by drawing a connection between AL enforcing message passing and social influence maximization, ensuring that the selected samples effectively improve the model performance. Second, we propose an extension to the influence model that incorporates an explicit *quality factor* to model label noise. In this way, we derive a fundamentally new AL selection criterion for GCN and LP–*reliable influence maximization* (RIM)–by considering quantity and quality of influence simultaneously. Empirical studies on public datasets show that RIM significantly outperforms current AL methods in terms of accuracy and efficiency.

## 1 Introduction

Graphs are ubiquitous in the real world, such as social, academic, recommendation, and biological networks [38, 30, 31, 10, 28]. Unlike the independent and identically distributed (i.i.d) data, nodes are connected by edges in the graph. Due to the ability to capture the graph information, message passing is the core of many existing graph models assuming that labels and features vary smoothly over the edges of the graph. Particularly in Graph Convolutional Neural Network (GCN) [16], the feature of each node is propagated along edges and transformed through neural networks. In Label Propagation (LP) [25], node labels are propagated and aggregated along edges in the graph.

The message passing typically requires a large amount of labeled data to achieve satisfactory performance. However, labeling data, be it by specialists or crowd-sourcing, often consumes too much time and money. The process is also tedious and error-prone. As a result, it is desirable to achieve good classification results with labeled data that is both few and unreliable. Active Learning (AL) [1] is a promising strategy to tackle this challenge, which minimizes the labeling cost by prioritizing the selection of data in order to improve the model performance as much as possible. Unfortunately, conventional AL methods [3, 8, 20, 41, 21, 27] treat message passing and AL independently without *explicitly* considering their impact on each other. In this paper, we advocate that a better AL method should unify node selection and message passing towards minimizing labeling cost, and we make two contributions towards that end.

35th Conference on Neural Information Processing Systems (NeurIPS 2021).

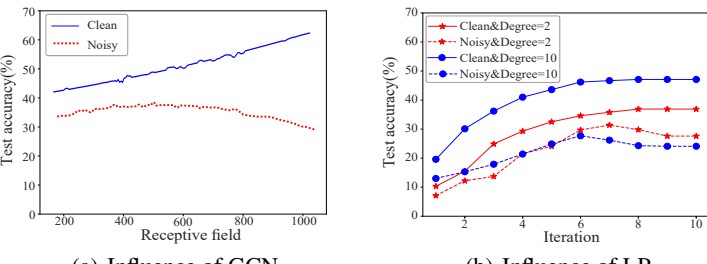

(a) Influence of GCN                    (b) Influence of LP

Figure 1: The influence between feature/label propagation and test accuracy with clean/noisy label.

The first contribution is that we quantify node influence by how much the initial feature/label of label node $v$ influences the output feature/label of node $u$ in GCN/LP, and then connect AL with influence maximization, e.g., the problem of finding a small set of seed nodes in a network that maximizes the spread of influence. To demonstrate this idea, we randomly select different sets of $|\mathcal{V}_l| = 20$ labeled nodes under the clean label and train a 2-layer GCN with a different labeled set on the Cora dataset [16]. For LP, we select two sets with the average node degree of 2 and 10. As shown in Figure 1, the model performance in both GCN/LP tends to increase along with the receptive field/node degree under the clean label, implying the potential gain of increasing the node influence.

Note that in real life, both humans and automated systems are prone to mistakes. To examine the impact of label noise, we set the label accuracy to 50%, and Figure 1(a) and Figure 1(b) show that the test accuracy could even drop with the increase of node influence under the noisy label. This is because the noise of labels will also be widely propagated with node influence increasing, thus diminishing the benefit of influence maximization. Therefore, our second contribution is that we further propose to maximize the reliable influence spread when label noise is taken into consideration. Specifically, each node is associated with a new parameter called the *quality* factor, indicating the probability that the label given by the oracle is correct. We recursively infer the quality of newly selected nodes based on the smoothing features/labels of previously selected nodes across the graph's edges, i.e., nodes that share similar features or graph structure are likely to have the same label.

Based on the above insights, we propose a fundamentally new AL selection criterion for GCN and LP–*reliable influence maximization* (RIM)–by considering both quantity and quality of influence simultaneously. Under a high-quality factor, we enforce the influence of selected label nodes for large overall reaches, while under a low-quality factor, we make mistake penalization to limit the selected node influence. RIM also maintains some nice properties such as submodularity, which allows a greedy approximation algorithm for maximizing reliable influence to reach an approximation ratio of $1 - \frac{1}{e}$ compared with the optimum. Empirical studies on public datasets demonstrate that RIM significantly outperforms the state-of-the-art methods GPA [13] by 2.2%-5.1% in terms of predictive accuracy when the labeling accuracy is 70%, even if it is enhanced with the anti-noise mechanism PTA [6]. Furthermore, in terms of efficiency, RIM achieves up to $4670\times$ and $18652\times$ end-to-end runtime speedups compared to GPA in GPU and CPU, respectively.

In summary, the core contributions of this paper are 1) We open up a novel perspective for efficient and effective AL for GCN and LP by enforcing the feature/label influence with a connection to *social influence maximization* [18, 9, 4]; 2) To the best of our knowledge, we are the first to consider the influence quality in graph-based AL, and we propose a new method to estimate the influence quantity based on the feature/label smoothing; 3) We combine the influence quality and quantity in a unified RIM framework. The empirical study on both GCNs and LP demonstrates that RIM significantly outperforms the compared baselines in performance and efficiency.

## 2 Preliminary

### 2.1 Active Learning

**Problem Formulation.** Let $c$ be the number of label classes and the ground-truth label for a node $v_i$ be a one-hot vector $\boldsymbol{y}_i \in \mathbb{R}^c$. Suppose that the entire node set $\mathcal{V}$ is partitioned into training set $\mathcal{V}_{train}$

(including both the labeled set $\mathcal{V}_l$ and unlabeled set $\mathcal{V}_u$), validation set $\mathcal{V}_{val}$ and test set $\mathcal{V}_{test}$. Given a graph $\mathcal{G} = (\mathcal{V}, \mathcal{E})$ with $|\mathcal{V}| = N$ nodes and $|\mathcal{E}| = M$ edges, feature matrix $\mathbf{X} = \{\boldsymbol{x_1}, \boldsymbol{x_2}..., \boldsymbol{x_N}\}$ in which $\boldsymbol{x_i} \in \mathbb{R}^d$, a labeling budget $\mathcal{B}$, and a loss function $\ell$, the goal of an AL algorithm is to select a subset of nodes $\mathcal{V}_l \subset \mathcal{V}_{train}$ to label from a noisy oracle with the labeling accuracy $\alpha$, so that it can produce a model $f$ with the lowest loss on the test set:

$$\underset{\mathcal{V}_l : |\mathcal{V}_l| = \mathcal{B}}{\arg\min} \ \mathbb{E}_{v_i \in \mathcal{V}_{test}} \left[ \ell \left( \boldsymbol{y}_i, P(\hat{\boldsymbol{y}}_i | f) \right) \right], \tag{1}$$

where $P(\hat{\boldsymbol{y}}_i | f)$ is the predictive label distribution of node $v_i$. To measure the influence of feature or label propagation, we focus on $f$ being GCN or LP.

**AL on graphs.** Both GCN and LP are representative semi-supervised graph models which can utilize additional unlabeled data to enhance the model learning [40], and lots of AL methods are specially designed for these two models. For example, both AGE [3] and ANRMAB [8] adopt uncertainty, density, and node degree when composing query strategies. LP-ME chooses the node that maximizes entropy for itself, while LP-MRE chooses the node such that the acquisition of its label reduces entropy most for the total graph [20].

## 2.2 Graph Models

**LP.** Based on the intuitive assumption that locally connected nodes are likely to have the same label, LP iteratively propagates the label influence to distant nodes along the edges as follows:

$$\mathbf{Y}^{(k+1)} = \widetilde{\mathbf{D}}^{-1} \widetilde{\mathbf{A}} \mathbf{Y}^{(k)}, \quad \mathbf{Y}_u^{(k+1)} = \mathbf{Y}^{(k+1)}, \quad \mathbf{Y}_l^{(k+1)} = \mathbf{Y}^{(0)}, \tag{2}$$

where $\mathbf{Y}^{(0)} = \{\boldsymbol{y_1}, \boldsymbol{y_2}..., \boldsymbol{y_l}\}$ is the initial label matrix consisting of one-hot label indicator vectors, $\mathbf{Y}_u^{(k)}$ and $\mathbf{Y}_l^{(k)}$ denote the soft label matrix of the labeled nodes set $\mathcal{V}_l$ and unlabeled nodes set $\mathcal{V}_u$ in iteration $k$ respectively. According to [39], we iteratively set the nodes in $\mathcal{V}_l$ back to their initial label $\mathbf{Y}^{(0)}$ since their labels are correct and should not be influenced by the unlabeled nodes.

**GCN.** Each node in GCN iteratively propagates its feature influence to the adjacent nodes when predicting a label. Especially, each layer updates the node feature embedding in the graph by aggregating the features of neighboring nodes:

$$\mathbf{X}^{(k+1)} = \delta \left( \widetilde{\mathbf{D}}^{-1} \widetilde{\mathbf{A}} \mathbf{X}^{(k)} \mathbf{W}^{(k)} \right), \tag{3}$$

where $\mathbf{X}^{(k)}$ and $\mathbf{X}^{(k+1)}$ are the embeddings of layer $k$ and $k + 1$ respectively. Specifically, $\mathbf{X}$ (and $\mathbf{X}^{(0)}$) is the original node feature. $\widetilde{\mathbf{A}} = \mathbf{A} + \mathbf{I}_N$ is used to aggregate feature vectors of adjacent nodes, where $\mathbf{A}$ is the adjacent matrix of the graph and $\mathbf{I}_N$ is the identity matrix. $\widetilde{\mathbf{D}}$ is the diagonal node degree matrix used to normalize $\widetilde{\mathbf{A}}$, $\mathbf{W}^{(k)}$ is a layer-specific trainable weight matrix and $\delta(\cdot)$ is the activation function.

## 2.3 Influence Maximization

The influence maximization (IM) problem in social networks aims to select $\mathcal{B}$ nodes so that the number of nodes activated (or influenced) in the social networks is maximized [15]. That being said, given a graph $\mathcal{G} = (\mathcal{V}, \mathcal{E})$, the formulation is as follows:

$$\underset{S}{\max} \ |\sigma(S)|, \ \mathbf{s.t.} \ S \subseteq \mathcal{V}, \ |S| = \mathcal{B}, \tag{4}$$

where $\sigma(S)$ is the set of nodes activated by the seed set $S$ under certain influence propagation models, such as Linear Threshold (LT) and Independent Cascade (IC) models [15]. The maximization of $\sigma(S)$ is NP-hard. However, if $\sigma(S)$ is nondecreasing and submodular with respect to $S$, a greedy algorithm can provide an approximation guarantee of $\left(1 - \frac{1}{e}\right)$ [23]. RIM is the first to connect social influence maximization with graph-based AL under a noisy oracle by defining reliable influence.

# 3 Reliable Influence Maximization

This section presents RIM, the first graph-based AL framework that considers both the influence quality and influence quantity. At each batch of node selection, RIM first measures the proposed reliable influence quantity and selects a batch of nodes that can maximize the number of activated nodes, and then it updates the influence quality for the next iteration. The above process is repeated until the labeling budget $\mathcal{B}$ runs out. We will introduce each component of RIM in detail below.

## 3.1 Influence Propagation

We measure the feature/label influence of a node $v_i$ on $v_j$ by how much change in the input feature/label of $v_i$ affects the *aggregated* feature/label of $v_j$ after $k$ iterations propagation [26, 33].

**Definition 3.1** (**Feature Influence**). *The feature influence score of node $v_i$ on node $v_j$ after k-step propagation is the L1-norm of the expected Jacobian matrix $\hat{I}_f(v_j, v_i, k) = \left\| \mathbb{E}[\partial \mathbf{X}_j^{(k)}/\partial \mathbf{X}_i^{(0)}] \right\|_1$. The normalized influence score is defined as*

$$I_f(v_j, v_i, k) = \frac{\hat{I}_f(v_j, v_i, k)}{\sum_{v_w \in \mathcal{V}} I(v_j, v_w, k)}. \tag{5}$$

**Definition 3.2** (**Label Influence**). *The label influence score of node $v_i$ on node $v_j$ after k-step propagation is the gradient of $\boldsymbol{y}_j^{(k)}$ with respect to $\boldsymbol{y}_i$:*

$$I_l(v_j, v_i, k) = \left\| \mathbb{E}[\partial \boldsymbol{y}_j^{(k)}/\partial \boldsymbol{y}_i] \right\|_1. \tag{6}$$

Given the $k$-step feature/label propagation mechanism, the feature/label influence score captures the sum over probabilities of all possible paths of length $k$ from $v_i$ to $v_j$, which is the probability that a random walk (or variants) starting at $v_j$ ends at $v_i$ after taking $k$ steps [33]. Thus, we could take them as the probability that $v_i$ propagates its feature/label to $v_j$ via random walks from the influence propagation perspective.

## 3.2 Influence Quality Estimation

Most AL works end up fully trusting the few available labels. However, both humans and automated systems are prone to mistakes (i.e., noisy oracles). Thus, we further estimate the label reliability and associate it with influence quality. The intuition behind our method is to exploit the assumption that labels and features vary smoothly over the edges of the graph; in other words, nodes that are close in feature space and graph structure are likely to have the same label. So we can recursively infer the newly selected node's quality based on the features/labels of previously selected nodes.

Specifically, after $k$ iterations of propagation, we calculate the node similarity $s$ of node $v_i$ and $v_j$ in LP by measuring the cosine similarity in $\mathbf{Y}^{(k)}$ according to E.q. (2). Like SGC [29], we remove the activate function $\delta(\cdot)$ and the trainable weight $\mathbf{W}^{(k)}$ and get the new feature matrix as: $\hat{\mathbf{X}}^{(k+1)} = \widetilde{\mathbf{D}}^{-1}\widetilde{\mathbf{A}}\hat{\mathbf{X}}^{(k)}$. For better efficiency in measuring the node similarity, we replace $\mathbf{X}^{(k)}$ in E.q. (3) with the simplified $\hat{\mathbf{X}}^{(k)}$ since its calculation is model-free. Similar to LP, we get the node similarity $s$ in GCN by measuring the cosine similarity in $\hat{\mathbf{X}}^{(k)}$. Larger $s$ means $v_i$ and $v_j$ are more similar after $k$ iterations of feature/label propagation and thus are more likely to have the same label.

**Theorem 3.1** (**Label Reliability**). *Denote the number of classes as c. And assume that the label is wrong with the probability of $1 - \alpha$, and the wrong label is picked uniformly at random from the remaining $c - 1$ classes. Given the labeled node $v_i \in \mathcal{V}_l$ and unlabeled node $v_j \in \mathcal{V}_u$, supposing the oracle labels $v_i$ as $\tilde{\boldsymbol{y}}_j$ and $\tilde{\boldsymbol{y}}_j = \boldsymbol{y}_i$ (the ground truth label for $v_i$, the same notation also applies to $v_j$), the reliability of node $v_j$ according to $v_i$ is*

$$r_{v_i \to v_j} = \frac{\alpha s}{\alpha s + (1 - \alpha)\frac{1-s}{c-1}} \tag{7}$$

*where $\alpha$ is the labeling accuracy, and $s$ measures the similarity between $v_i$ and $v_j$.*

Proof of Theorem 3.1 is in Appendix A.1. Intuitively, Theorem 3.1 shows that the label of node $v_j$ is more reliable if (1) The oracle is more proficient and thus has higher labeling accuracy $\alpha$. (2) The labeled node $v_i$ is more similar to $v_j$ and thus leads to larger $s$.

**Definition 3.3** (**Influence Quality**). *The influence quality of $v_j$ is recursively defined as*

$$r_{v_j} = \sum_{v_i \in \mathcal{V}_l, \tilde{\boldsymbol{y}}_j = \tilde{\boldsymbol{y}}_i} \hat{r}_{v_i} r_{v_i \to v_j}, \tag{8}$$

*where $r_{v_i \to v_j}$ is the label reliability of node $v_j$ with respect to node $v_i$, and $\hat{r}_{v_i} = \frac{r_{v_i}}{\sum_{v_q \in \mathcal{V}_l, \tilde{\boldsymbol{y}}_j = \boldsymbol{y}_q} r_{v_q}}$ is the normalized label reliability score of node $v_i$.*

For each unlabeled node $v_j$, we firstly find all labeled nodes which have the same label with $\tilde{\boldsymbol{y}}_j$ and then get the final influence quality with weighted voting [17]. The source node $v_i$ should contribute more in measuring its influence on another node $v_j$ if its label is reliable, i.e., with higher $\hat{r}_{v_i}$.

## 3.3   Reliable Influence

Different from the original social influence method which only considers the influence magnitude [14, 34], we measure the influence quantity and get the reliable influence quantity by introducing the influence quality since it is common to have noisy oracles in the labeling process of Active Learning.

**Definition 3.4** (**Reliable Influence Quantity Score**). *Given the influence quality $r_{v_i}$, the reliable influence quantity score of node $v_i$ on node $v_j$ after k-step feature/label propagation is*

$$Q(v_j, v_i, k) = r_{v_i} I(v_j, v_i, k), \tag{9}$$

*where $I(v_j, v_i, k)$ is $I_f(v_j, v_i, k)$ for GCN and $I_l(v_j, v_i, k)$ for LP.*

The reliable influence quantity score $Q(v_j, v_i, k)$ is determined by: (1) The influence quality of the labeled influence source node $v_i$. (2) The feature/label influence score of $v_i$ on $v_j$ after k-step propagation. From the perspective of random walk, it is harder to get a path from $v_i$ to $v_j$ with larger steps $k$, and the influence score will gradually decay along with the propagation steps. As a result, node $v_j$ can get a highly reliable influence quality score from the labeled node $v_i$ if they are close neighbors and $v_i$ has high influence quality.

In a node classification problem, the node label is dominated by the maximum class in its predicted label distribution. Motivated by this, we assume an unlabeled node $v_j$ can be activated if and only if the maximum influence magnitude satisfies:

$$Q(v_j, \mathcal{V}_l, k) > \theta, \tag{10}$$

where $Q(v_j, \mathcal{V}_l, k) = \max_{v_i \in \mathcal{V}_l} Q(v_j, v_i, k)$ is the maximum influence of $\mathcal{V}_l$ on the node $v_j$, and the threshold $\theta$ is a parameter which should be specially tuned for a given dataset.

**Definition 3.5** (**Activated Nodes**). *Given a set of labeled seeds $\mathcal{V}_l$, the activated node set $\sigma(\mathcal{V}_l)$ is a subset of nodes in $\mathcal{V}$ that can be activated by $\mathcal{V}_l$:*

$$\sigma(\mathcal{V}_l) = \bigcup_{v \in \mathcal{V}, Q(v, \mathcal{V}_l, k) > \theta} \{v\}. \tag{11}$$

The threshold $\theta = 0$ means we consider an unlabeled node $v$ is influenced as long as there is a $k$-step path from any labeled node, which is equal to measuring whether this unlabeled node can be involved in the entire training process of GCN or LP. In practice, we could choose this threshold value in the case of a tiny budget so that our goal is to involve unlabeled nodes as more as possible. However, if the budget is relatively large, we could choose a positive threshold $\theta > 0$, which enables the selection process to pay more attention to those weakly influenced nodes.

## 3.4   Node Selection and Model Training

In order to increase the feature/label influence (smoothness) effect on the graph, we should select nodes that can influence more unlabeled nodes. Due to the impact of the graph structure, the speed

---

**Algorithm 1:** Batch Node Selection.

---

**Input:** Initial labeled set $\mathcal{V}_0$, query batch size $b$, and labeling accuracy $\alpha$.
**Output:** Labeled set $\mathcal{V}_l$

1   $\mathcal{V}_l = \mathcal{V}_0$;
2   **for** $t = 1, 2, \ldots, b$ **do**
3      Select the most valuable node $v^* = \arg\max_{v \in \mathcal{V}_{train} \setminus \mathcal{V}_l} \ F(\mathcal{V}_l \cup \{v\})$;
4      Set the influence quality of $v^*$ to the labeling accuracy $\alpha$;
5      Update the labeled set $\mathcal{V}_l = \mathcal{V}_l \cup \{v^*\}$;
6   Update the influence quality of nodes in $\mathcal{V}_l \setminus \mathcal{V}_0$ according to E.q. 8;
7   **return** $\mathcal{V}_l$

---

of expansion or, equivalently, growth of the influence can change dramatically given different sets of label nodes. This observation motivates us to address the graph data selection problem in the viewpoint of influence maximization defined below.

**RIM Objective.** Specifically, RIM adopts a reliable influence maximization objective:

$$\max_{\mathcal{V}_l} F(\mathcal{V}_l) = |\sigma(\mathcal{V}_l)|, \textbf{s.t. } \mathcal{V}_l \subseteq \mathcal{V}, \ |\mathcal{V}_l| = \mathcal{B}. \tag{12}$$

By considering both the influence quality and quantity, RIM aims to find a subset of $\mathcal{B}$ to label so that the number of activated nodes can be maximized.

**Reliable Node Selection.** Without losing generality, we consider a batch setting where $b$ nodes are selected in each iteration. For the first batch, when the initial labeled set $\mathcal{V}_0 = \emptyset$, we ignore the influence quality term $r_{v_i}$ in E.q. 9 since there are no reference nodes for measuring the label quality from the oracle. For better efficiency in finding selected nodes in each batch, we set the influence quality of each node to the labeling accuracy $\alpha$ during the node selection process and then simultaneously update these values according to E.q. 8 after the node selection process of this batch. For the node selection after the first batch, Algorithm 1 provides a sketch of our greedy selection method for the graph models, including both GCN and LP. Given the initial labeled set $\mathcal{V}_0$, query batch size $b$, and labeling accuracy $\alpha$, we first select the node $v^*$ generating the maximum marginal gain (line 3), set its influence quality to the labeling accuracy $\alpha$ (line 4), and then update the labeled set $\mathcal{V}_l$ (line 5). After getting a batch of labeled nodes, we require the label from a noisy oracle (line 5) and then update their influence quality according to E.q. 8 (line 6).

**Theorem 3.2.** *The greedily selected batch node set is within a factor of $(1 - \frac{1}{e})$ of the optimal set for the objective of reliable influence maximization.*

Proof of Theorem 3.2 is in Appendix A.2. The node selection strategy in both AGE and ANRMAB relies on the model prediction, while this process in RIM is model-free. Such a characteristic is helpful in practice since the oracle does not need to wait for the end of model training in each batch node selection of RIM. For example, if the model training dominates the end-to-end runtime, the main efficiency overhead of both AGE and ANRMAB is the model training.

**Reliable Model Training.** Nodes with larger influence quality in $\mathcal{V}_l$ should contribute more to the training process, so we introduce the influence quality in the model training. For GCN, we use the weighted cross entropy loss as: $\mathcal{L} = -\sum_{v_i \in \mathcal{V}_l} r_{v_i} \boldsymbol{y_i} \log \hat{\boldsymbol{y}}_i$, where $r_{v_i}$ is the influence quality of node $v_i$. For each labeled node $v_i$ in LP, we just change its label $\boldsymbol{y_i}$ to $r_{v_i} \boldsymbol{y_i}$.

### 3.5   Comparison with Prior Works

Existing AL works [35, 7, 2, 5] regarding label noise mainly focus on two phases: noise detection and noise handling. Both the data-driven and model-based methods are designed for noise detection. The former firstly constructs a graph from the dataset and then utilizes graph properties, e.g., the homophily assumption [22] without corrupting graph structure [24], while the latter [35] measures the likelihood of noise by the predicted soft labels [32]. For noise handling, existing works primarily concentrate on three aspects: data correction, objective function modification, and optimization policy modification [12]. However, none of these AL methods is specially designed for graphs and fails to consider the influence quantity imposed by the graph structure, leading to sub-optimal performance.

Meanwhile, there are also works [37, 8, 36] concerning graph-based AL, and they are designed for GCN or LP. Nonetheless, they fail to take into account the noise brought by oracles. That being said, the quality of influence is overlooked. Thus these methods suffer from a lack of robustness, especially when the quantity of noise is substantial. To sum up, current AL methods are unable to consider the quantity and quality of influence simultaneously. To bridge this gap, RIM introduces the influence quality to tackle the noise from the oracle. Besides, the influence quantity has been deliberated to get more nodes involved in semi-supervised learning.

## 4    Experiments

We now verify the effectiveness of RIM on four real-world graphs. We aim to answer four questions. **Q1:** Compared with other state-of-the-art baselines, can RIM achieve better predictive accuracy? **Q2:** How does the influence quality and quantity influence RIM? **Q3:** Is RIM faster than the compared baselines in the end-to-end AL process? **Q4:** If RIM is more effective than the baselines, what should be the reason?

### 4.1    Experiment Setup

**Datasets and Baselines.** We use node classification tasks to evaluate RIM in both inductive and transductive settings [11] on three citation networks (i.e., Citeseer, Cora, and PubMed) [16] and one large social network (Reddit). The properties of these datasets are summarized in Appendix A.3. We compare RIM with the following baselines: (1) **Random**: Randomly select the nodes to query; (2) **AGE** [3]: Combine different query strategies linearly with time-sensitive parameters for GCN; (3)**ANRMAB** [8]: Adopt a multi-armed bandit mechanism for adaptive decision making to select nodes for GCNs; (4)**GPA** [13]: Jointly train on several source graphs and learn a transferable active learning policy which can directly generalize to unlabeled target graphs; (5)**LP-MRE** [20]: Select a node that can reduce the entropy most in LP; (6)**LP-ME** [20]: Select a node with maximum uncertainty score in LP.

None of these baselines has considered the label noise in AL. For a fair comparison, we have tried several anti-noise mechanisms [12, 24, 32] to fight against noise in GCN and LP, and finally choose PTA [6] to our baselines since it can get the best performance in most datasets. We name AGE enhanced with PTA as AGE+, so do other baselines. Similar to RIM, PTA assigns each labeled node a dynamic label reliability score for model training. PTA computes the label reliability based on the graph proximity and the similarity of the predicted label, while RIM does not consider the model prediction because it may be unreliable in the AL setting that only a few labeled nodes can be used in the initial node selection phases. Unlike PTA, RIM considers the labeling accuracy and combines it with the graph proximity in its influence quality estimation.

**Implementations.** We use OpenBox [19] for hyper-parameter tuning or follow the original papers to find the optimal hyperparameters for each method. To eliminate randomness, we repeat each method ten times and report the mean test accuracy. The implementation details are shown in Appendix A.4, and our code is available in the supplementary material.

### 4.2    Experimental Results

**Performance on GCN.** To answer **Q1**, We choose the labeling size as 20 nodes per class labeling error rate ranging from 0 to 0.5, and then we report the corresponding test accuracy of GCN in Figure 2. Compared to other baselines, RIM consistently outperforms the baselines as the labeling error rate grows. Moreover, even with the anti-noise method, i.e., PTA, the baselines still have a noticeable performance gap from RIM. To demonstrate the improvement of RIM in the noisy situation, we also provide the test accuracy with a labeling error rate set as 0.3. Table 1 shows that GPA+, AGE+, and ANRMAB+ outperform Random in most datasets, as they are specially designed for GCNs. However, RIM further boosts the performance by a significant margin. RIM improves the test accuracy of the best baseline, i.e., GPA+, by 3.5-5.1% on the three citation networks and 2.2% on the Reddit dataset.

**Performance on LP.** Following the same settings in GCN, the result of LP is shown in Figure 3. Even if combined with the anti-noise method PTA, RIM consistently outperforms LP-MRE and

Table 1: The test accuracy (%) on different datasets when labeling accuracy is 0.7.

| Model | Methods | Cora | Citeseer | PubMed | Reddit |
|-------|---------|------|----------|--------|--------|
| GCN | Random | 65.6 | 56.3 | 63.3 | 75.2 |
| | AGE+ | 72.5 | 61.1 | 68.3 | 77.6 |
| | ANRMAB+ | 72.4 | 63.4 | 68.9 | 77.2 |
| | GPA+ | 72.8 | 63.8 | 69.7 | 77.9 |
| | RIM | **77.9** | **67.5** | **73.2** | **80.1** |
| LP | Random | 51.7 | 31.4 | 50.4 | 51.3 |
| | LP-ME+ | 55.7 | 35 | 56.1 | 53.4 |
| | LP-MRE+ | 59.1 | 41.4 | 58.5 | 54.9 |
| | RIM | **62.4** | **46.7** | **65.5** | **58.5** |

Table 2: The influence of different components in RIM.

| Method | Cora | Δ | Citeseer | Δ | PubMed | Δ |
|--------|------|------|----------|------|--------|------|
| No RT | 75.1 | -2.8 | 63.9 | -3.6 | 71.4 | -1.8 |
| No RS | 74.8 | -3.1 | 63.4 | -4.1 | 70.5 | -2.7 |
| No RTS | 73.4 | -4.5 | 61.9 | -5.6 | 68.9 | -4.3 |
| **RIM** | **77.9** | – | **67.5** | – | **73.2** | – |

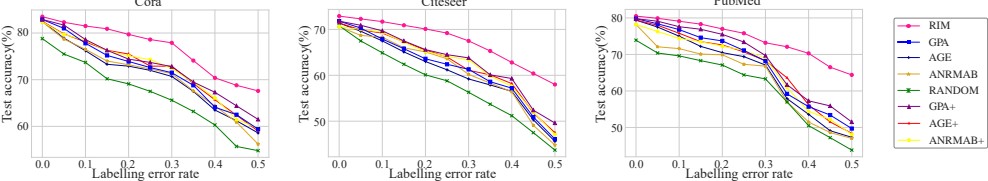

Figure 2: The test accuracy with different labeling error rate of labeled nodes for GCN.

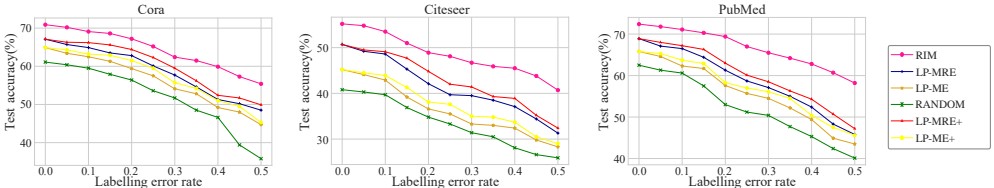

Figure 3: The test accuracy with different labeling error rate of labeled nodes for LP.

LP-ME by a large margin as the labeling error rate grows. To demonstrate the improvement of RIM with the existence of noise, we also provide the test accuracy with labeling accuracy set as 0.7. Table 1 shows that both LP-ME+ and LP-MRE+ outperform Random, but RIM further boosts the performance by a significant margin. Especially, RIM improves test accuracy of the best baseline LP-MRE+ by 3.3-7.0% on the three citation networks and 3.6% on Reddit.

**Ablation Study.** RIM combines both influence quality and influence quantity. To answer **Q2** and verify the necessity of each component, we evaluate RIM on GCN while disabling one component at a time when the labeling accuracy is 0.7. We evaluate RIM: *(i)* without the label reliability score served as the loss weight (called "No Reliable Training (RT)"); *(ii)* without the label reliability when selecting the node (called "No Reliable Selection (RS)");*(iii)* without both reliable component(called "No Reliable Training and Selection (RTS)"). Table 2 displays the results of these three settings.

The influence quality in RIM contributes to both the reliable node selection and reliable training. First, the test accuracy will decrease in all three datasets if reliable training is ignored. For example, the performance gap is as large as 3.6% if reliable training is unused on Citeseer. Adopting reliable training can avoid the bad influence from the nodes with low label reliability. Besides, reliable node selection has a significant impact on model performance on all datasets, and it is more important than reliable training since removing the former will lead to a more substantial performance gap. For example, the gap on PubMed is 2.7%, which is higher than the other gap (1.8%). The more reliable the label is, the more reliable activated nodes we can use to train a GCN.

With the removal of reliable training and selection, the objective of RIM is to maximize the influence quantity (the total number of activated nodes). As shown in Table 1 and 2, RIM still exceeds the AGE+ method by a margin of 0.9%, 0.8%, and 0.6% on Cora, Citeseer, and PubMed, respectively, which verifies the effectiveness of maximizing the influence quantity.

**Influence of labeling budget** We study the performance of different AL methods under different labeling budgets in order to answer **Q1**. More concretely, the budget size ranges from $2k$ to $20k$ with labeling error rates being 0 and 0.3, respectively, and report the test accuracy of GCN. Figure 4 shows

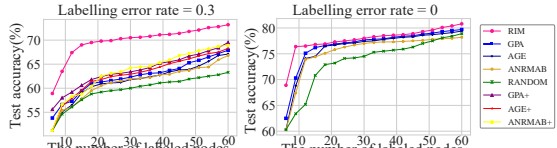
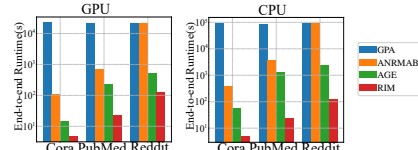

Figure 4: The test accuracy of each method along with different labeling budget when labeling error rate is 0.3 and 0, respectively.

Figure 5: The end-to-end runtime (at log scale) on different datasets. The speedup on each bar is relative to GPA.

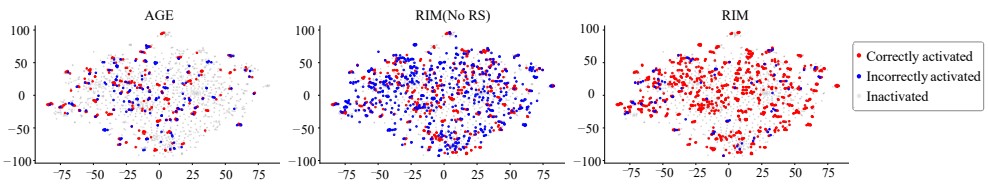

Figure 6: The node distribution of different methods on the Cora dataset. Note that a node is correctly activated only if it is firstly activated by a correctly labelled node.

that with the budget size growing, RIM constantly outperforms other baselines, especially when there is more label noise, which shows the robustness of RIM.

**Efficiency Comparison.** Another key advantage of RIM is its high efficiency in node selection. To answer **Q3**, we report the end-to-end runtime of each method in Figure 5. In real-world AL settings, the end-to-end runtime of model-based methods (i.e., GPA, AGE, and ANRMAB) includes both labeling time and model training time. However, the labeling time is excluded from our measurement since the quality and proficiency of oracles dramatically influence it.

The left part in Figure 5 shows that RIM obtains a speedup of 22×, 30×, and 172× over ANRMAB on Cora, PubMed, and Reddit and a speedup of 4670×, 950×, and 177× over GPA on Cora, PubMed, and Reddit, respectively on GPUs. It is worth mentioning that, since GPA is a transfer learning method, the training time does not depend on the scale of the target dataset, and thus the runtime of GPA is very close on the three datasets (its details can be found in Appendix A.4). As RIM is model-free and does not rely on GPU, we also evaluate them on CPU environment, and the result is shown in the right part of Figure 5. The speedup grows on all these three datasets. For example, the speedup increases from 4670x to 18652x when the GPU device is unavailable on the Cora dataset.

**Interpretability.** To answer **Q4**, we evaluate the distribution of the correctly activated nodes, incorrectly activated nodes, and inactivated nodes for AGE, RIM (No RS), and RIM when labeling accuracy is 0.7 for Cora in GCN. The result in Figure 6 shows that AGE has the fewest activated nodes, and nearly half of them are incorrectly activated. RIM (No RS) has the most activated nodes but also gets many nodes activated by incorrectly labeled nodes. Compared to these two methods, RIM has enough activated nodes, and most of them are activated by correctly labeled nodes (i.e., restrain the noisy propagation), which is why RIM performs better in node classification.

## 5   Conclusion

Both GCN and LP are representative graph models which rely on feature/label propagation. Efficient and effective data selection for the model training is demanding due to its inherent complexity, especially in the real world when the oracle provides noisy labels. In this paper, we propose RIM, a novel AL method that connects node selection with social influence maximization. RIM represents a critical step in this direction by showing the feasibility and the potential of such a connection. To accomplish this, we firstly introduce the concept of feature/label influence and then define their influence quality/quantity. To deal with the oracle noise, we propose a novel criterion to measure the influence quality based on the graph isomorphism. Finally, we connect the influence quality with the influence quantity and propose a new objective that maximizes the reliable influence quantity.

Empirical studies on real-world graphs show that RIM outperforms competitive baselines by a large margin in terms of both model performance and efficiency. We are extending RIM to heterogeneous graphs for future work as the current measurement of influence quality cannot be directly used.

## Broader Impact

Specifically, RIM can be employed in graph-related areas such as prediction on citation networks, social networks, chemical compounds, transaction graphs, road networks, etc. Each of the usage may bring a broad range of societal benefits. For example, predicting the malicious accounts on transaction networks can help identify criminal behaviors such as stealing money and money laundry. Prediction on road networks can help to avoid traffic overload and saving people's time. RIM has significant technical-economic and social benefits because it can significantly shorten the labor time and labor intensity of oracles. However, as RIM requires oracles to label each selected node, it also faces the risk of information leakage. In this regard, we encourage researchers to understand the privacy concerns of RIM and investigate how to mitigate possible information leakage.

## Acknowledgments and Disclosure of Funding

This work is supported by NSFC (No. 61832001, 61972004), Apple Scholars in AI/ML PhD fellowship, Beijing Academy of Artificial Intelligence (BAAI), PKU-Baidu Fund 2019BD006, and PKU-Tencent Joint Research Lab. Zhi Yang and Bin Cui are the corresponding authors.

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
