# RIM: Reliable Influence-based Active Learning on Graphs

**Wentao Zhang**[1,2]**, Yexin Wang**[1]**, Zhenbang You**[1]**, Meng Cao**[2]**, Ping Huang**[2]
**Jiulong Shan**[2]**, Zhi Yang**[1,3]**, Bin Cui**[1,3,4]
[1]School of CS, Peking University [2]Apple
[3] National Engineering Laboratory for Big Data Analysis and Applications
[4]Institute of Computational Social Science, Peking University (Qingdao), China
[1]{wentao.zhang, yexinwang, zhenbangyou, liyang.cs, yangzhi, bin.cui}@pku.edu.cn
[2]{mengcao, Huang_ping, jlshan}@apple.com

## A  Appendix

### A.1  Proof of Theorem 1

**Theorem 1** (**Label Reliability**). *Denote the number of classes as c. And assume that the label is wrong with the probability of $1 - \alpha$, and the wrong label is picked uniformly at random from the remaining $c - 1$ classes. Given the labelled node $v_i \in \mathcal{V}_l$ and unlabelled node $v_j \in \mathcal{V}_u$, suppose the oracle labels $v_j$ as $\tilde{\boldsymbol{y}}_j$ and $\tilde{\boldsymbol{y}}_j = \boldsymbol{y}_i$ (the ground truth label for $v_i$, the same notation also applies to $v_j$), the reliability of node $v_j$ according to $v_i$ is*

$$r_{v_i \to v_j} = \frac{\alpha s}{\alpha s + (1 - \alpha)\frac{1-s}{c-1}} \tag{1}$$

*where $\alpha$ is the labelling accuracy, and $s$ is the probability that $v_i$ and $v_j$ actually have the same label.*

In practice, we can estimate $\alpha$ with redundant votes across oracles (e.g., such as Amazon's Mechanical Turk) by treating the majority vote as correct labels, like the Dawid-Skene algorithm [2].

Mathematically speaking, what we want is a conditional probability. To be more precise, we have already know that the label for $v_j$ given by the oracle is the same as the ground truth label of $v_i$, and we want to calculate the probability of the event that the label for $v_j$ given by the oracle is the same as the ground truth label of $v_j$. Formally, the reliability of node $v_j$ according to $v_i$ is

$$r_{v_i \to v_j} = Pr\left\{\tilde{\boldsymbol{y}}_j = \boldsymbol{y}_j | \tilde{\boldsymbol{y}}_j = \boldsymbol{y}_i\right\}, \tag{2}$$

With the definition of conditional probability, we have

$$Pr\left\{\tilde{\boldsymbol{y}}_j = \boldsymbol{y}_j | \tilde{\boldsymbol{y}}_j = \boldsymbol{y}_i\right\} = \frac{Pr\left\{\tilde{\boldsymbol{y}}_j = \boldsymbol{y}_j, \tilde{\boldsymbol{y}}_j = \boldsymbol{y}_i\right\}}{Pr\left\{\tilde{\boldsymbol{y}}_j = \boldsymbol{y}_i\right\}} = \frac{Pr\left\{\tilde{\boldsymbol{y}}_j = \boldsymbol{y}_j, \boldsymbol{y}_j = \boldsymbol{y}_i\right\}}{Pr\left\{\tilde{\boldsymbol{y}}_j = \boldsymbol{y}_i\right\}} \tag{3}$$

Then we shall calculate the numerator and denominator, respectively.

For denominator, with the law of total probability, we have

$$Pr\left\{\tilde{\boldsymbol{y}}_j = \boldsymbol{y}_i\right\} = Pr\left\{\tilde{\boldsymbol{y}}_j = \boldsymbol{y}_i, \boldsymbol{y}_j = \boldsymbol{y}_i\right\} + Pr\left\{\tilde{\boldsymbol{y}}_j = \boldsymbol{y}_i, \boldsymbol{y}_j \neq \boldsymbol{y}_i\right\} \tag{4}$$

Then calculate these two terms separately.

$$\begin{aligned} Pr\left\{\tilde{\boldsymbol{y}}_j = \boldsymbol{y}_i, \boldsymbol{y}_j = \boldsymbol{y}_i\right\} &= Pr\left\{\tilde{\boldsymbol{y}}_j = \boldsymbol{y}_j, \boldsymbol{y}_j = \boldsymbol{y}_i\right\} \\ &= Pr\left\{\tilde{\boldsymbol{y}}_j = \boldsymbol{y}_j\right\} \cdot Pr\left\{\boldsymbol{y}_j = \boldsymbol{y}_i\right\} = \alpha s \end{aligned} \tag{5}$$

35th Conference on Neural Information Processing Systems (NeurIPS 2021).

Table 1: Overview of the Four Datasets

| Dataset | #Nodes | #Features | #Edges | #Classes | #Train/Val/Test | Task type | Description |
|---|---|---|---|---|---|---|---|
| Cora | 2,708 | 1,433 | 5,429 | 7 | 1,208/500/1,000 | Transductive | citation network |
| Citeseer | 3,327 | 3,703 | 4,732 | 6 | 1,827/500/1,000 | Transductive | citation network |
| Pubmed | 19,717 | 500 | 44,338 | 3 | 18,217/500/1,000 | Transductive | citation network |
| Reddit | 232,965 | 602 | 11,606,919 | 41 | 155,310/23,297/54,358 | Inductive | social network |

The correctness of the second equal sign is due to the independence of the correctness of the oracle and the conformity of ground truth labels of two i.i.d. samples.

$$Pr\left\{\tilde{\boldsymbol{y}}_j = \boldsymbol{y}_i, \boldsymbol{y}_j \neq \boldsymbol{y}_i\right\} = Pr\left\{\tilde{\boldsymbol{y}}_j = \boldsymbol{y}_i | \boldsymbol{y}_j \neq \boldsymbol{y}_i\right\} \cdot Pr\left\{\boldsymbol{y}_j \neq \boldsymbol{y}_i\right\} = \frac{1-\alpha}{c-1} \cdot (1-s) \quad (6)$$

Add them and the denominator is solved,

$$Pr\left\{\tilde{\boldsymbol{y}}_j = \boldsymbol{y}_i\right\} = \alpha s + \frac{1-\alpha}{c-1} \cdot (1-s) \tag{7}$$

Now calculate the numerator.

$$Pr\left\{\tilde{\boldsymbol{y}}_j = \boldsymbol{y}_j, \boldsymbol{y}_j = \boldsymbol{y}_i\right\} = Pr\left\{\tilde{\boldsymbol{y}}_j = \boldsymbol{y}_j\right\} \cdot Pr\left\{\boldsymbol{y}_j = \boldsymbol{y}_i\right\} = \alpha s \tag{8}$$

Then we have the final answer,

$$r_{v_i \to v_j} = \frac{\alpha s}{\alpha s + \frac{1-\alpha}{c-1}(1-s)} \tag{9}$$

Therefore, the theorem follows.

## A.2 Proof of Theorem 2

**Definition 1** (**Nondecreasing submodular**). *Given a set $S$, and the function $F(\cdot)$, $|F(S)|$ is nondecreasing submodular with respect to $S$ if $\forall S \subseteq T, v \notin T, |F(T)| \geq |F(S)|$ and $|F(S \cup \{v\})| - |F(S)| \geq |F(T \cup \{v\})| - |F(T)|$.*

Previous work [8] shows a greedy algorithm can provide an approximation guarantee of $(1 - \frac{1}{e})$ if $|F(S)|$ is nondecreasing and submodular with respect to $S$.

Consider a batch setting with $\frac{\mathcal{B}}{b}$ rounds where $b$ nodes are selected in each iteration (see Algorithm 1). Theorem 3.2 states that the greedy selection returns a $(1 - \frac{1}{e})$-approximate to the RIM objective for each batch selection, i.e., $\max_{\mathcal{V}_b} F(\mathcal{V}_b) = |\sigma(\mathcal{V}_l \cup \mathcal{V}_b)|$, s.t. $\mathcal{V}_b \subseteq \mathcal{V} \setminus \mathcal{V}_l$, $|\mathcal{V}_b| = b$, where $\mathcal{V}_l$ is the set of nodes selected in previous rounds.

We can prove $F$ is submodular as follows:

For every $A \subseteq B \subseteq S$ and $s \in S \setminus B$, let $Q_A(v) = \max_{v_i \in \mathcal{V}_l \cup A} Q(v, v_i, k)$ and $Q_B(v) = \max_{v_j \in \mathcal{V}_l \cup B} Q(v, v_j, k)$. Since $(\mathcal{V}_l \cup A) \subseteq (\mathcal{V}_l \cup B)$, for any $v \in \mathcal{V}$, $Q_A(v) \leq Q_B(v)$, so we have:

$F(A \cup \{s\}) - F(A) = |\{v \mid Q(v, s, k) > \theta \geq Q_A(v)\}| \geq |\{v \mid Q(v, s, k) > \theta \geq Q_B(v)\}| = F(B \cup \{s\}) - F(B)$

Therefore, the Theorem follows.

## A.3 Dataset description

**Cora**, **Citeseer**, and **Pubmed**[1] are three popular citation network datasets, and we follow the public training/validation/test split in GCN [7]. In these three networks, papers from different topics are considered as nodes, and the edges are citations among the papers. The node attributes are binary word vectors, and class labels are the topics papers belong to.

---

[1] https://github.com/tkipf/gcn/tree/master/gcn/data

**Reddit** is a social network dataset derived from the community structure of numerous Reddit posts. It is a well-known inductive training dataset, and the training/validation/test split in our experiment is the same as that in GraphSAGE [4]. The public version provided by GraphSAINT[2] [13] is used in our paper. For more specifications about the four aforementioned datasets, see Table 1.

**ogbn-arxiv** is a directed graph, representing the citation network among all Computer Science (CS) arXiv papers indexed by MAG. The training/validation/test split in our experiment is the same as the public version. The public version provided by OGB[3] is used in our paper.

**ogbn-papers100M** is a paper citation dataset with 111 million papers indexed by MAG [11] in it. This dataset is currently the largest existing public node classification dataset and is much larger than others. We follow the official training/validation/test split and metric released in the official website[4] and official paper [6].

## A.4 Implementation details

For Cora and Citeseer, the threshold $\theta$ is chosen as 0.05, while for PubMed and Reddit, the threshold $\theta$ is chosen as 0.005.

In terms of GPA [5], so as to obtain its full performance, the pre-trained model released by its authors on Github is adopted. More precisely, for Cora, we choose the model pre-trained on PubMed and Citeseer; for PubMed, we choose the model pre-trained on Cora and Citeseer; for Citeseer and Reddit, we choose the model pre-trained on Cora and PubMed. Other hyper-parameters are all consistent with the released code.

When it comes to AGE [1] and ANRMAB [3], in order to obtain well-trained models and guarantee that the model-based selection criteria employed by them run well, GCN is trained for 200 epochs in each node selection iteration. For LP [10], the number of propagation iterations is set to 10. AGE is implemented with its open-source version and ANRMAB in accordance with its original paper.

In addition, $c$ (i.e., the number of classes) nodes are chosen to be labeled in each iteration. As an instance, $c$ is chosen as 7 in Cora.

Efficiency measurement is carried out on each of the four datasets. Models are all trained for 2000 epochs to measure the end-to-end runtime. It is worth noting that the runtime of GPA on all four datasets is virtually identical, e.g., the end-to-end runtime with GPU on Cora, PubMed, Citeseer, and Reddit is 22,416s, 21,983s, 22,175s, and 22,319s, respectively, which can be justified by the fact that the RL model of GPA is trained on small datasets, whereas its time complexity is irrelevant to the scale of datasets.

The experiments are conducted on an Ubuntu 16.04 system with Intel(R) Xeon(R) CPU E5-2650 v4 @ 2.20GHz, 4 NVIDIA GeForce GTX 1080 Ti GPUs and 256 GB DRAM. All the experiments are implemented in Python 3.6 with Pytorch 1.7.1 [9] on CUDA 10.1.

## A.5 Experiments on OGB datasets

To verify the effectiveness of RIM on large graphs, we add the experiments on ogbn-arxiv and ogbn-papers100M. Due to the large memory cost, GCN cannot be implemented on ogbn-papers100M in a single machine, thus we use the simplified GCN [12] to replace the original GCN here [7].

The experimental results in Table 2 shows that RIM has better performance and robustness than other baselines in these two datasets. Note that it takes more than one week for model-based baselines to finish the AL process on the large ogbn-papers100M, and we mark these methods as out-of-time(OOT).

---

[2] https://github.com/GraphSAINT/GraphSAINT
[3] https://ogb.stanford.edu/docs/nodeprop/#ogbn-arxiv
[4] https://github.com/snap-stanford/ogb

Table 2: The test accuracy (%) on different ogb datasets when labeling accuracy is 0.7.

| Model | Methods | ogbn-arxiv | ogbn-paper100M |
|-------|---------|------------|----------------|
| SGC | Random | 47.7 | 44.6 |
| | AGE+ | 53.9 | OOT |
| | ANRMAB+ | 54.1 | OOT |
| | GPA+ | 56.3 | OOT |
| | RIM | **60.8** | **48.7** |
| LP | Random | 42.6 | 39.1 |
| | LP-ME+ | 47.2 | 39.9 |
| | LP-MRE+ | 51.3 | OOT |
| | RIM | **54.9** | **44.3** |