# OpenReview forum: "RIM: Reliable Influence-based Active Learning on Graphs"
_NeurIPS.cc/2021/Conference — NeurIPS 2021 Spotlight_

### Official Review · Reviewer_nYPR · 2021-07-10

**Rating:** 6
**Confidence:** 3

**Summary:**

This paper studies graph active learning in the presence of labeling noise, which is an under-explored area with real-world impact. The key idea of this paper is to define Reliable-Influence-Quantity-Score Q as the product of label reliability and label influence. The active learning objective is then to maximize the number of activated node, whose Q score exceed a certain threshold. The authors proposed a greedy algorithm for selecting the nodes, and showed the greedy algorithm is close to optimal. Empirical results on real-world dataset shows this algorithm outperforms the baselines by large margins.

**Limitations And Societal Impact:**

Limitation:
1. The setup of the paper is not 100% clear to me. In particular, there are three type of labels: 1) the ground truth, 2) labels given by an oracle, 3) the predicted labels given by the algorithm. In the problem statement in section 2.1, it seems that the ground truth labels are not available, and we only have access to the labeling oracle. However, in equation 8, the ground truth labels are used for computation. From the content, it seems that there are at least a small set of nodes where the ground truth label are given, I hope the authors can further clarify the setup in their writing.
2. In appendix line 14, s is defined as the probability that the two node i,j have the same ground truth label. In practice, s is computed as the cosine similarity between the two node features / or label vectors. This seems to be a huge leap since: 1) cosine similarity has range [-1,1] while probability has range [0,1]; 2) even if those two have the same range, simply using cosine similar as probability without further justification is still not sound enough.
3. It is not clear to me how equation 6 in appendix is derived. In particular, where does k come from?

Societal Impact:
There is no significant negative societal impact associated with this research.

**Main Review:**

The problem consider in this paper is of practical impact. The motivation of the paper is clearly presented. The paper seem to be well written overall, but part of the problem setup is not 100% clear (see next). The method proposed in this paper is both inspiring and gives good empirical gains. I think this paper should be accepted after some moderated concerns being addressed. I will be more confident in my judgement after my questions (see next) are addressed.

**Time Spent Reviewing:**

5

---

> ### Author Response · Authors · 2021-08-10
> **Response to Reviewer #4**
>
> Thank you for your positive review of our submission and valuable feedback. We address each of your questions below.
>
> ### Setup
> Sorry for the unclear setup due to the notation error. The reviewer is right that the ground truth labels are not available, and we only have access to the labels given by oracles. We find a notation error in Eq.(8) and correct $\\small \\boldsymbol{y}\_i$ as $\\small \\tilde{\\boldsymbol{y}}\_i$: $\\small r\_{v\_j} = \\frac{1}{|\\{v\_i\\in \\mathcal{V}\_l|\\tilde{\\boldsymbol{y}}\_j = \\boldsymbol{y}\_i\\}|}\\sum\_{v\_i\\in \\mathcal{V}\_l, \\tilde{\\boldsymbol{y}}\_j = \\boldsymbol{y}\_i } r\_{v\_i \\rightarrow v\_j} \\approx \\sum\_{v\_i\\in \\mathcal{V}\_l, \\tilde{\\boldsymbol{y}}\_j =\\tilde{ \\boldsymbol{y}}\_i } \\hat{r}\_{v\_i}r\_{v\_i \\rightarrow v\_j}$. The intuition behind is that $\\small \\tilde{\\boldsymbol{y}}\_j = \\boldsymbol{y}\_i$ might not be directly observed due to the unknown ground-truth labels $\\small \\boldsymbol{y}\_i$, so we approximate it by $\\small \\tilde{\\boldsymbol{y}}\_j =\\tilde{ \\boldsymbol{y}}\_i$ and adopt bias towards node $\\small i$ with larger $\\hat{r}\_{v\_i}$ to enforce good approximation, as larger $\\small \\hat{r}\_{v\_i}$ indicates a high likelihood of being $\\small \\tilde{ \\boldsymbol{y}}\_i = \\boldsymbol{y}\_i$.
>
> ### Computation of $s$
> The reviewer raises an excellent point. To avoid the range problem, we guarantee that the cosine similarity of node features/label vectors is between 0 and 1 by applying MinMaxScaler [Komer et al. 2014] to normalize the feature vector to be non-negative (elements of the label vector are all non-negative). Since the features of all the experimented datasets (Cora, Citeseer, PubMed, and Reddit) are non-negative, we do not apply such normalization in our experiment.
> Our experiment shows that using similarity to indicate probability $\\small s$ can indeed work well, which leverages the common assumption of GCN and LPA: connected nodes tend to have the same ground-truth label. By smoothing features of nodes across the edges of the graph, we make the features of nodes with the same label similar and thus using feature similarity between nodes to indicate the $\\small s$. In fact, a more sophisticated method could be introduced to estimate probability $\\small s$ based on similarity. For example, we could conduct a K-means cluster based on the similarity and get the label (cluster) probability distribution of each node, based on which we could derive $\\small s$ (i.e., the probability of finding a pair of nodes have the same label).
>
> [Komer et al. 2014] Brent Komer, James Bergstra, and Chris Eliasmith. [Hyperopt-sklearn: automatic hyperparameter configuration for scikit-learn.](https://conference.scipy.org/proceedings/scipy2014/pdfs/komer.pdf) ICML workshop on AutoML, 2014.
>
> ### Eq.6 derivation
> Thanks for bringing this to our attention. As in our response to Reviewer 1, we will explicitly state the assumption in the proof: with probability $\\small 1-\\alpha$, the label is wrong, and the wrong label is picked uniformly at random from the remaining $\\small k-1$ classes. Here, $\\small k$ is the number of classes for a given dataset. So in Eq.6 in the proof, $\\small Pr\\{\\tilde{y}\_j=y\_i|y\_j\\neq y\_i\\} = \\frac{1-\\alpha}{k-1}$ under the above uniform assumption.

---

### Official Review · Reviewer_RKEJ · 2021-07-14

**Rating:** 7
**Confidence:** 2

**Summary:**

This paper proposes the AL method which combines node selection and message passing of the graph models for minimizing labeling cost. The first technical contribution is to quantify the influence of labeling in GCN/LP and relate AL to the problem of finding seed nodes for maximizing the influence. In addition, this paper shows how to maximize the spread of influence by considering the noise and reliability of labels.

**Limitations And Societal Impact:**

I think the applicability of the proposed technology is not low, but there is little description of its social value and impact. It would be better to add descriptions of specific applications that would inspire readers (users of the technology).

**Main Review:**

- This paper is well structured, clearly written and easy-to-follow. The idea of the method seems to me to be highly convincing and reasonable. In particular, the method that takes into account the noise of the labels seems to have a high practical value.
- It has been shown to be effective in terms of accuracy and computational complexity for node classification tasks in cases where there is noise in the labels. It has also been validated using several real data sets, and comparison results with many baselines are provided.
- The authors used PTA as the anti-noise mechanisms. What other algorithms have you tried, and why did you choose just PTA? It is better to show that the proposed method is superior without relying on the selection of anti-noise mechanisms.

**Time Spent Reviewing:**

3 hours

---

> ### Author Response · Authors · 2021-08-10
> **Response to Reviewer #3**
>
> ## Response to Reviewer-3
> Thanks for your positive review of our submission and valuable feedback.
>
> ### The usage of PTA
> We have discussed it in Line 250-251:  "For a fair comparison, we have tried a number of anti-noise mechanisms provided in [9,20,25] to fight against noise in GCN and LP, and finally choose PTA for a *strong baseline* since it can get the better performance than other mechanisms in most datasets."
>
>
> ### Societal value and impact
> Specifically, RIM can be employed in graph-related areas such as prediction on citation networks, social networks, chemical compounds, transaction graphs, road networks, etc. Each of the usage may bring a broad range of societal benefits. For example, predicting the malicious accounts on transaction networks can help identify criminal behaviors such as stealing money and money laundry. Prediction on road networks can help to avoid traffic overload and saving people’s time. RIM has significant technical-economic and social benefits because it can significantly shorten the labor time and labor intensity of oracles. However, as RIM requires oracles to label each selected node, it also faces the risk of information leakage. In this regard, we encourage researchers to understand the privacy concerns of RIM and investigate how to mitigate possible information leakage.
>
> We will add the discussion of societal impact in our final version.

---

> > ### Comment · Reviewer_RKEJ · 2021-09-01
> > **Thank you for your responses**
> >
> > I agree with adding the above discussion to the final version.

---

### Official Review · Reviewer_e5Eo · 2021-07-16

**Rating:** 6
**Confidence:** 4

**Summary:**

This proposes a novel graph-based active learning framework by reliable social influence maximization. The proposed model aims to select a subset of nodes to be labeled, so as to maximize the number of nodes influenced. Whether a node is influenced is determined by both the influence quantity and quality (assuming label noise). A greedy algorithm is adopted for optimization, whose approximation precision is guaranteed theoretically. SeveraleExperiments are conducted to validate the effectiveness and efficiency of RIM.

**Limitations And Societal Impact:**

The authors adequately addressed the limitations and potential negative societal impact of their work.

**Main Review:**

This paper proposes an interesting active learning method on graph models from a social influence maximization perspective. It exploits the message passing property of graph models, quantifying the influence quantity of labeled nodes to unlabeled nodes, so as to incorporate active learning into the message passing process. In addition, label noise is considered, and label reliability is achieved by estimating influence quality. The overall method is theoretically sound, theorem 3.1 obtains label reliability, and theorem 3.2 guarantees the feasibility of Algorithm 1.

In general, this paper is well-organized, clearly written, and easy to follow. The experiments are relatively sufficient and demonstrate the effectiveness of RIM. Ablation study supports the necessity of influence quality. The interpretability illustration is impressive. However, I have the following questions and concerns:
(1) The experiments do not demonstrate the advantage of influence quantity; the claim/motivation of “A better AL method should unify node selection and message passing” is not well supported. Table 2 shows “No RTS” does not always outperform the baselines, e.g., GPA+ on Citeseer and PubMed. It will be nice to conduct an additional verification by removing the effects from label noise, e.g., to compare “No RTS” with AL baselines(no anti-noise mechanism), such as GPA, in a setting without label noise.

(2) What’s the relationship between r_{v_j} in Equation 8 and Q(v_j,v_i,k) in Equation 9? It seems that both the two capture the reliability of v_j. But Q(v_j,v_i,k) is used to decide whether v_j is activated, and r_{v_j} is used during model training. Why?

**Time Spent Reviewing:**

5 hours

---

> ### Author Response · Authors · 2021-08-10
> **Response to Reviewer #2**
>
> Thank you for your review and valuable feedback.
>
> ### Influence quantity verification
> Note that the RIM objective is reduced to only maximizing the influence quantity in the case without noise. We show this case in (1) Figure 2 and Figure 3, where the nodes with a labeling error rate of 0, (2) Figure 4(b) where there is no label noise. These figures show that RIM outperforms baselines and demonstrate the advantage of influence quantity. In fact, the above results also provide a verification of removing the effects from label noise (in Table 2) suggested by the reviewer, as “No RTS” is only to maximize the influence quantity by removing the quality-based component.
>
> ### Relationship between Eq.8 and Eq.9
> $\\small r\_{v\_j}$ measures the influence *quality* of node $\\small v\_j$ (i.e., the likelihood that the oracle label of $v\_j$ is correct), whereas $\\small Q(v\_j,v\_i,k)$ in Eq.9 further considers the influence *quantity* $\\small I(v\_j,v\_i,k)$ (i.e., how much the change in the input feature/label of $\\small v\_i$ affects the feature/label of $\\small v\_j$ via propagation) besides the quality $\\small r\_{v\_j}$. We use $\\small Q(v\_j,v\_i,k)$ to decide whether $\\small v\_j$ is activated because it consider the effects of both propagation (quantity) and noise (quality).
> Additionally, we use $\\small r\_{v\_j}$ during the model training, which is inspired by recent works on confidence-aware loss functions. We clarify that the $\\small r\_{v\_j}$ here is used for determining the sample confidence and contribution to training. Besides the model training, $\\small r\_{v\_j}$ is also used for node selection in Eq.9.

---

> ### Author Response · Authors · 2021-09-02
> **Response to Reviewer #2**
>
> Thanks for your helpful and insightful reviews.
>
> As shown in both Figure 2 and Figure 4(b) of the original paper, we have already demonstrated the advantage of influence quantity in a setting without label noise.
> Besides,  we have also carefully explained the relationship between Eq.8 and Eq.9 in our previous response.
>
> We hope our response can solve the problems you proposed, and we are happy to respond if some problems still exist.
>
> Respectfully,
>
> Paper1228 Authors

---

### Official Review · Reviewer_qTkm · 2021-07-16

**Rating:** 7
**Confidence:** 4

**Summary:**

The authors propose a novel graph-based active learning approach, specifically with label propagation and graph convolutional networks in mind. They, in particular, tackle the problem of label noise in graph-based active learning, which is indeed very important and not well studied. They provide a conceptual and practical solution to this problem. For that, they introduce the notion of reliable influence maximization based on influence maximization in social networks, where they take not only the quantity of the influence into account but also a novel way of measuring the influence quality. On benchmark datasets, they achieve impressive performance in terms of predictive accuracy, runtime and label noise robustness compared to some baselines.

**Limitations And Societal Impact:**

The authors do not discuss limitations of their work, but that their approach is not directly applicable to heterogeneous networks (line 334,335).

 In real-world applications, the labeling accuracy $\alpha$ is usually unknown. The authors could add a discussion on what to do in such situations.

The authors motivate their approach in Eq. (1) with the minimization of the generalization risk. However, they later in the paper do not refer to it again. Is is possible to relate the author's RIM optimization to the actual goal of minimizing the risk. Are there reasonable conditions/conditions to achieve that?

They also do not discuss any societal impact, but this is fine, as their work is an early conceptual / methodological work.

**Main Review:**

Even though the general idea has much potential, there are serious issues with the writing (e.g., notation, language, clarity, references),  missing references, and many missing details in the theory and experimental section.

*** If the authors properly address the following issues, I am more than willing to change my rating. ***

The theoretical derivations seem correct but miss many details and do not state all assumptions. Most importantly, the proof of the key theoretical contribution of this paper - that the (reliable influence maximization) RIM objective is submodular - is discussed very shortly in the appendix. A rigorous proof is necessary here.

Additionally, in the proof of Theorem 1, the last step in Eq. (6) is only possible by assuming that the label noise is happening in a particular way: with probability $1-\alpha$, the label is wrong (this is what the authors state) and that the wrong label is picked uniformly at random from the remaining $k-1$ classes (this is not explicitly assumed or stated in the paper). Also, it is not completely clear whether the ground-truth labels (e.g., $y_i$) are fixed (deterministic) or rather random variables. In the main text, it seems like they are fixed (and only the $\tilde{y}$ are random), but in the proof, they use the $y_i$ as random variables (e.g., state the probability $P(y_i = y_j)$). Clarification helps a lot here. It is also not the Bayes formula in line 11, but merely the definition of conditional probabilities.

It is also unclear how the authors actually use Theorem 1 because it assumes the true similarity/probability s to be known, while the authors use an estimate of the similarity given by LP (label propagation) or GCN (graph convolution networks) instead. I also would disagree, that the computation of RIM is "model-free" because it still relies on evaluating, e.g., LP some number of steps. Could the authors help disambiguate and clarify the issues here?

It is also not completely clear why there is an expectation in the definition of feature and label influence. In the paper [23], the expectation is only used in the feature influence and not in the label influence. The authors of [26] use the same definition without the expectation. I can only speculate that it is either the expectation over the randomly sampled features $X$ and/or labels $y$, or meant in a social influence maximization manner (with, e.g., probabilities that vertex v influences vertex w). How are these expectations computed?

More things are not clear and not well explained: The modification of the cross-entropy loss and the labels using the influence quality ("reliable model training") is not explained nor motivated well and might even bias in favour of the proposed method. The motivation with Figure 1 is not really clear (also: what exactly do you mean with "receptive field").

Empirical evaluations:
The achieved results definitely are impressive, in particular, in noisy settings. The code seems to be reproducible and well documented.
However, there are multiple issues.

1. Even though the achieved mean accuracy of 10 runs is significantly higher than the baselines, error bars would make the plots more valuable. The authors claimed that they have error bars in the Checklist, which they do not; also not in the supplementary.

2. The four used datasets (especially the three citation datasets) are generally known to be overused in the last years.
    Other graphs should have been used instead or additionally.

3. It is unclear why the authors selected these particular baselines. For example, they have chosen a (comparably unknown) variant of active LP [17], instead of the (first) classical active LP (and one of the most known graph active learning) approach [Zhu, 2003]. The other baselines are: AGE [3] (an unpublished preprint from 2017), [10] (2018) and a recent approach from last year. There exist many well-tested graph-based active learning approaches such as [Ma, 2013] and [Gu, 2012], including the baselines therein. In fact, [Ma, 2013] seem to achieve comparable accuracy on Cora and even better accuracy on Citeseer using LP without noise (comparing Figure 3 of [Ma, 2013] with the current paper's Figure 3). Can the authors explain their choice of methods?

4. The authors say that they used grid-search to select hyperparameters. However, more details on how, for example, their activation threshold $\theta$ was chosen are missing. Experiments on how $\theta$ affects the results are not shown and would benefit the experimental evaluation.

5. Looking at Figure 4: It is not mentioned on what dataset these plots were achieved. Why not present such plots for all datasets (e.g., in the supplementary)? Also, it looks like the proposed method has already a ~7% better test accuracy than all baselines after only (roughly) 5 queries, which seems to be too good to be true.

6. Looking at Figure 5: Why did the author's exclude the baselines LP-ME and LP MRE here? In line 304, they say that Figure 5 contains the "runtime of each method". The reason to exclude the labeling time in these measurements is not clear to me (line 306/307). They also mention in line 196 that to increase the efficiency, they set the influence of each new vertex first to $\alpha$. Plots showing how this affects the efficiency (but also the accuracy) are interesting.

7. It seems like RIM is the only algorithm with direct access to the label accuracy $\alpha$, while the baselines do not explicitly use this information. A discussion about this discrepancy is necessary.

8. How do the approaches select the first vertex (or batch $V_0$) to query?

Missing / not discussed important previous work:

* In line 66/67 the authors claim that they are the first to consider label noise in graph-based active learning (AL). However, there are previous graph AL approaches that also include label noise. For example, [Dasarathy, 2015] explicitly models the label noise as a parameter and derives bounds based on it. Also, multiple approaches model graph-based AL in a probabilistic setting (e.g., Gaussian random fields) with probability distributions for each vertex to have a particular label, e.g, the original active LP [Zhu, 2003] and [Ma, 2013]. This also models label noise but just not with an explicit parameter $\alpha$ as the authors of this submission did.

* The use of submodularity in AL and connections between influence maximization and AL: The use of submodularity is quite common in AL, see, e.g., [Chen 2013], [Ma, 2013]. In fact, the whole subject of "adaptive submodularity" [Golovin 2013] deals with the connection between submodularity and AL (and also social influence maximization).

(Easily fixable) notation issues:

* $k$ is the number of classes (/labels) (e.g., line 74, proof of Theorem 1 and the Theorem itself), but also the number of iterations (e.g., eq. 2, 3. def 3.1, 3.2, 3.4, 3.5).

* no difference between $s$ the estimated (by LP or GCN) similarity of two vertices and $s$ the probability that two vertices have the same ground truth label.

* The "global" budget B and the batch size b should be related, but this is not discussed. In general, I think the additional abstraction of using a batch of size b instead of just querying B vertices iteratively, is unnecessary and does not contribute much. It instead makes the understanding of Algorithm 1 more difficult.

* The authors sometimes call the influence function $F$ and sometimes $\sigma$. This is especially troubling in Def. 1 (nondecreasing submodular) in the appendix, where they switch the notation in the middle of the equation. In eq. (12) "RIM objective" it is also unnecessary to write max $F(.) = |\sigma(.)|$

* Why is there an underline in line 205 and also in line 33 of the appendix.

* Looking at the most left equation in Eq. 2, it seems like Y^k and Y^(k+1) is the full |V|*k vertex label matrix. In the other two equation however, the same matrix is assigned to Y_l and Y_u of different size.

Some smaller language issues:
Line 219 "The former .. construct" -> "constructs". In the input of Alg.1, the authors say "labelling" while on other occasions, they say "labeling" (e.g., line 202). Please, be consistent.

Some issues in the references:

Why do the authors cite [13] and [14] the preprint and published versions of GCN? Reference [21] contains no authors and no title but only the editors and the conference; is the intention to cite the conference as a whole? Citation [15] seems to be wrong. The authors cite it under the keyword "social influence maximization" in line 66 (btw. I do not understand what "[15]; 2)" there means), but [15] is about a different topic. They might confuse it with [Chen 2013]. Sometimes the authors spell out the same conference name, e.g., [24] sometimes not, e.g., [26]. Please, be consistent.

References:

[Zhu, 2003]: Zhu, Xiaojin, John Lafferty, and Zoubin Ghahramani. "Combining active learning and semi-supervised learning using gaussian fields and harmonic functions." ICML 2003 workshop on the continuum from labeled to unlabeled data in machine learning and data mining 2003.

[Ma, 2013] Ma, Yifei, Roman Garnett, and Jeff G. Schneider. "Σ-Optimality for Active Learning on Gaussian Random Fields." NIPS. 2013.

[Gu, 2012] Gu, Quanquan, and Jiawei Han. "Towards active learning on graphs: An error bound minimization approach." ICDM 2012.

[Dasarathy, 2015] Dasarathy, G., Nowak, R., & Zhu, X. (2015, June). S2: An efficient graph based active learning algorithm with application to nonparametric classification. In Conference on Learning Theory

[Chen 2013] Chen, Yuxin, and Andreas Krause. "Near-optimal batch mode active learning and adaptive submodular optimization." ICML 2013.

[Golovin 2013] Golovin, Daniel, and Andreas Krause. "Adaptive submodularity: Theory and applications in active learning and stochastic optimization." Journal of Artificial Intelligence Research 42 (2011): 427-486.

**Time Spent Reviewing:**

12

---

> ### Author Response · Authors · 2021-08-10
> **Response to Reviewer #1, Part 1**
>
> Thanks for your careful review and feedback!  All the comments are helpful.
> ## 1. Theory
>
> ### Submodular derivation
> We will add a rigorous proof. We consider a batch setting with $\\small \\mathcal{B}/b$ rounds where $\\small b$ nodes are selected in each iteration (see Alg.1). Besides, Theorem 3.2 states that the greedy selection returns a $\\small (1-\\frac{1}{e})$-approximation to the RIM objective for each batch selection, i.e.,$\\small \\max\_{\\mathcal{V}\_b}F(\\mathcal{V}\_b)= |\\sigma(\\mathcal{V}\_l \\cup \\mathcal{V}\_b)|, \\mathbf{s.t.}\\ {\\mathcal{V}\_b}\\subseteq \\mathcal{V}\\setminus \\mathcal{V}\_l,\\ |{\\mathcal{V}\_b}|=b$, where $\\mathcal{V}\_l$ is the set of nodes selected in all previous rounds. We can prove that $\\small F$ is submodular: For every $\\small A\\subseteq B \\subseteq S$ and $\\small s \\in S \\setminus B$, let $\\small Q\_A(v) =\\max\_{v\_i\\in \\mathcal{V}\_l \\cup A} Q(v,v\_i,k)$ and $\\small Q\_B(v) =\\max\_{v\_j\\in \\mathcal{V}\_l \\cup B} Q(v,v\_j,k)$. Clearly, for any $\\small v\\in \\mathcal{V}$, $\\small Q\_A(v) \\leq Q\_B(v)$, and we have:
>
> $\\small F(A\\cup\\{s\\})-F(A)=  \\left|\\left\\{v\\mid Q(v,s,k) > \\theta \\geq Q\_A(v)\\right\\}\\right|\\geq \\left|\\left\\{v\\mid Q(v,s,k) > \\theta \\geq Q\_B(v)\\right\\}\\right| = F(B\\cup\\{s\\})-F(B)$
>
> ### Assumption statement
> In the proof of Theorem 1, the reviewer is right that we make the standard uniform assumption as previous works [8]: the label is wrong with the probability of $\\small 1-\\alpha$, and the wrong label is picked uniformly at random from the remaining $\\small k-1$ classes. We will explicitly state this assumption in Theorem 1. Note that our analysis in the proof could also be applied to more general cases where the label error distribution $\\small Pr\\{\\tilde{y}\_j=y\_i|y\_j\\neq y\_i\\}$ is given.
>
> ### Ground-truth labels
> Thank you for bringing this to our attention. We find a notation error in Eq.(8) and correct $\\small \\boldsymbol{y}\_i$ as $\\small \\tilde{\\boldsymbol{y}}\_i$: $\\small r\_{v\_j} = \\frac{1}{|\\{v\_i\\in \\mathcal{V}\_l|\\tilde{\\boldsymbol{y}}\_j = \\boldsymbol{y}\_i\\}|} \\sum\_{v\_i\\in \\mathcal{V}\_l, \\tilde{\\boldsymbol{y}}\_j = \\boldsymbol{y}\_i } r\_{v\_i \\rightarrow v\_j}  \\approx \\sum\_{v\_i\\in \\mathcal{V}\_l, \\tilde{\\boldsymbol{y}}\_j =\\tilde{ \\boldsymbol{y}}\_i } \\hat{r}\_{v\_i}r\_{v\_i \\rightarrow v\_j}$. The intuition behind is that $\\small \\tilde{\\boldsymbol{y}}\_j = \\boldsymbol{y}\_i$ cannot be observed due to the unknown ground-truth labels $\\small \\boldsymbol{y}\_i$, so we approximate it by $\\small \\tilde{\\boldsymbol{y}}\_j =\\tilde{ \\boldsymbol{y}}\_i$ and adopt bias towards node $\\small i$ with larger $\\small \\hat{r}\_{v\_i}$ to enforce good approximation, as larger $\\small \\hat{r}\_{v\_i}$ indicates higher likelihood of being $\\small \\tilde{ \\boldsymbol{y}}\_i = \\boldsymbol{y}\_i$.
> After this notation correction, we clarify that we treat both Oracle labels (e.g., $\\small \\tilde{ \\boldsymbol{y}}\_j$) and ground-truth labels (e.g., $\\small \\boldsymbol{y}\_i$) as variables when modeling the conditional probability $\\small r\_{v\_i \\rightarrow v\_j} = Pr\\{\\tilde{\\boldsymbol{y}}\_j=\\boldsymbol{y}\_j|\\tilde{\\boldsymbol{y}}\_j=\\boldsymbol{y}\_i\\}$ in Theorem 3.1 and the proof. In Eq.8, we treat Oracle-label *observations* as fixed to find conditional events (e.g., $\\small \\tilde{ \\boldsymbol{y}}\_j=\\tilde{\\boldsymbol{y}}\_i$) for the quality computation. We will clarify this with different notations for variables (e.g., $\\small \\tilde{ \\boldsymbol{y}}\_i$) and fixed observations  (e.g., $\\small \\tilde{\\boldsymbol{y}}\_i'$) of oracle labels. We also correct the line 11 in the proof as a conditional probability definition.
>
> ### The Use of Theorem 1
> The reviewer is right that Theorem 1 requires the true probability $\\small s=P(\\boldsymbol{y}\_i=\\boldsymbol{y}\_j)$. Since the ground-truth labels are unknown, we use the similarity of smoothed features to estimate $s$. This leverages the common assumption of GCN and LPA: connected nodes tend to have the same ground-truth label. By smoothing features of nodes across the edges of the graph, we make the features of nodes with the same label similar, thereby using feature similarity between nodes to estimate the $s$. More details about such estimation can be further found in our response to reviewer 4.
>
> ### Model-free
> We agree that ''model-free'' is ambiguous for LP. We will clarify it by restricting this term in the setting of AL for the learning-based model such as GCN, compared to the traditional learning-based AL methods with guidance from the trained model.
>
> ### Expectation in definition
> Since RIM decouples node selection and model training, both the feature $\\small \\mathbf{X}^{(k)}$ and the label $\\small y^{(k)}$ can be fixed, i.e., $\\small \\mathbb{E}[\\partial \\mathbf{X}\_j^{(k)}/\\partial \\mathbf{X}\_i^{(0)}] = \\partial \\mathbf{X}\_j^{(k)}/\\partial \\mathbf{X}\_i^{(0)}$ and $\\small \\mathbb{E}[\\partial\\boldsymbol{y}\_j^{(k)}/\\partial \\boldsymbol{y}\_i] = \\partial\\boldsymbol{y}\_j^{(k)}/\\partial \\boldsymbol{y}\_i$. In this sense, the reviewer is right that the expectation is not really needed. However, we still keep the expectation here as a more general definition of influence which allows extending the idea of RIM to learning-based methods. For example, we can use the feature embedding $\\small \\mathbf{X}^{(k)}$ (e.g., in GCN) or the label prediction $\\small \\mathbf{y}^{(k)}$ (e.g., in APPNP [Klicpera et al. 2019]) given by the model to compute influence, which are variables and need expectation.
>
> [Klicpera et al. 2019] Johannes Klicpera, Aleksandar Bojchevski, Stephan Günnemann. [Predict then Propagate: Graph Neural Networks meet Personalized PageRank](https://openreview.net/forum?id=H1gL-2A9Ym). ICLR 2019.
>
> ### Loss modification
> The idea is inspired by recent works on confidence-aware loss functions in noisy-label learning. The confidence score (weight) of each sample determines its contribution to training -- more confident instances are those with high influence quality scores and thus contribute more to training.
>
> ### The motivation with Figure 1
> Given a K-layer GCN, only nodes within the K-hop neighborhood of the labeled nodes affect the model training. The K-hop neighbors are referred to as the Receptive Field (RF) of the labeled nodes. Fig.1 highlights that involving more nodes during training can boost model performance, but the need to handle noise increases simultaneously, motivating our reliable influence concept combining both influence quantity (node involvement) and quality (noisy).

---

> > ### Author Response · Authors · 2021-08-10
> > **Response to Reviewer #1, Part 2**
> >
> > ## 2. Empirical evaluations
> >
> > ### Implementation details
> > We keep the same settings as before. Concretely, we choose the labeling budget as $\\small 20$ nodes per class for all datasets. The threshold $\\small \\theta$ is chosen as 0.05 for two small datasets: Cora and Citesser, and 0.005 for other datasets. To alleviate randomness, we repeat each method ten times.
> >
> >
> > | Model | Methods |     Cora     |   Citeseer   |    PubMed    |    Reddit    |  ogbn-arxiv  | ogbn-papers100M |
> > |:-----:|:-------:|:------------:|:------------:|:------------:|:------------:|:------------:|:---------------:|
> > |       | Random  |   65.6±2.7   |   56.32±.6   |   63.32±.6   |   75.2±3.4   |   47.7±3.8   |    44.6±4.3     |
> > |       |  AGE+   |   72.5±2.5   |   61.1±2.5   |   68.3±2.3   |   77.6±2.8   |   53.9±3.4   |       OOT       |
> > |  GCN  | ANRMAB+ |   72.4±2.5   |   63.4±2.4   |   68.9±2.2   |   77.2±2.9   |   54.1±3.5   |       OOT       |
> > |       |  GPA+   |   72.8±2.3   |   63.8±2.2   |   69.7±1.7   |   77.9±2.5   |   56.3±3.0   |       OOT       |
> > |       | **RIM** | **77.9±1.5** | **67.5±1.5** | **73.2±1.3** | **80.1±1.6** | **60.8±1.7** |  **48.7±2.4**   |
> >
> > | Model | Methods | Cora | Citeseer | PubMed | Reddit | ogbn-arxiv | ogbn-papers100M |
> > |:-----:|:-------:|:----:|:--------:|:------:|:------:|:----------:|:---------------:|
> > |       | Random  |   51.7±2.5   |   31.4±2.3   |   50.4±2.4   |   51.3±3.1   |   42.6±3.5   |    39.1±4.1     |
> > |  LP   | LP-ME+  |   55.7±2.2   |   35.0±2.2   |   56.1±2.0   |   53.4±2.7   |   47.2±2.8   |    39.9±3.3     |
> > |       | LP-MRE+ |   59.1±2.0   |   41.4±2.1   |   58.5±1.8   |   54.9±2.6   |   51.3±3.0   |       OOT       |
> > |       | **RIM** | **62.4±1.3** | **46.7±1.2** | **65.5±0.9** | **58.5±1.4** | **54.9±1.5** |  **44.3±2.1**   |
> >
> > ### Error bars.
> > Thanks for your suggestion, and we add the mean accuracy and corresponding standard deviation. As the above table shows, RIM has a lower standard deviation and thus is more robust in both GCN and LP than other baselines. The reason is that the standard deviation of RIM comes from different label distributions given by the oracle (i.e., the correctly labeled node may be given a wrong label at another time) and the training of the GCN model. Besides these two factors, the standard deviation of other model-based methods will also be influenced by randomly selected nodes in the first batch and the training of the GCN/LP model whose prediction is used for node selection. The standard deviation of GCN is higher than LP because the latter is parameter-free.
> >
> > ### Experiments on Other graphs.
> > Thanks for your suggestion, and we add the experiments on ogbn-arxiv and ogbn-papers100M. The above table shows that RIM has better performance and robustness than other baselines in all datasets. Note that it takes more than one week for model-based baselines to finish the AL process on the large ogbn-papers100M, and we mark these methods as out-of-time(OOT).
> >
> > ### The selection of baselines
> > While many graph-based AL approaches exist, they are ineffective to GCN since they cannot capture both graph structures and node features. AL for GCN is a newly emergent research topic. AGE (though unpublished) and ANRMAB are selected as baselines as they are two significant works focus on AL for GCN. As for LP, we use a variant of active LP because it can handle multi-classification problems, whereas classical LP is designed for binary classification. We thank the reviewer for reminding us of [Ma, 2013] and [Gu, 2012], and we will include these baselines. [Ma, 2013] reported better accuracy on Citeseer because it only used the largest connected component (2109 nodes), whereas we used the whole dataset (3327 nodes). This is also why the baseline Random in [Ma, 2013] has much better accuracy than that in our experiment. In fact, our primary purpose is to show that RIM can (1) support both GCN and LP and (2) can explicitly handle noise, rather than showing RIM can outperform all competing LP-based AL methods without noise.
> >
> > ### How $\\small \\theta$ affects the results
> > The parameter $\\small \\theta$ controls how easily a node is activated. We select the $\\small \\theta$ based on the given labeling budget: If the budget is small, the overall influence of the labeled node-set is relatively weak, and we set a small threshold (close to 0) to make the node more easily be activated. Otherwise, we set a larger threshold to ensure that the influence of $\\small \\mathcal{V}\_l$ on each activated node $v\_j$ is strong. To test the influence of $\\small \\theta$, we vary its value in [0.01, 0.05, 0.1] in Cora and get the corresponding test accuracy of [77.4%, 77.9% and 77.6%] respectively when the oracle accuracy is 0.7. As shown above, RIM is robust to the parameter $\\small \\theta$.
> >
> > ### Details of Figure 4
> > The experiment is conducted with GCN on PubMed. RIM gets a ~7% better test accuracy than all baselines after only five queries because other baselines are learning-based, which requires the predictions of trained models to guide node selection. Unfortunately, the prediction may be inaccurate in the initial AL iterations with few labels, especially in noisy settings. By contrast, our method is independent of the model, thus significantly better than those baselines in the first few queries. The results for other datasets are similar, and we will add them in the revised supplementary.
> >
> > ### Details of Figure 5
> > The figure plots the runtime of AL methods for GCN because the training of GCN is more time-consuming than LP due to the optimization of the model parameter. We do not consider the human-intensive oracle labeling time, which depends on the complexity of a specific task and the proficiency of the oracle.
> >
> > ### Access to $\\alpha$
> > Compared to existing works, RIM can directly utilize the potential of graph-based AL under the noisy oracle by *explicitly* maximizing the new concept of reliable influence with direct access to labeling accuracy $\\small \\alpha$.
> >
> > ### First batch
> > We set the influence quality of each node to the labeling accuracy $\\small \\alpha$, and use the greedy selection (Algorithm 1).
> >
> > ## 3. Related works
> >
> > ### Lable noise
> > For LP, the label noise has already been considered by many graph-based AL methods, such as [Dasarathy, 2015], the original active LP [Zhu, 2003], and [Ma, 2013]. However, RIM can also support AL for GCN with feature smoothing instead of LP. Also, RIM optimizes the node selection under the noisy oracle by explicitly exploiting the labeling accuracy $\\small \\alpha$.
> >
> > ### Connection
> > Although previous works connect submodularity with either AL or social influence maximization, RIM is the first to directly connect AL and social influence maximization with the proposed concept of reliable influence.
> >
> > ## 4. Notation and reference issues
> >
> > Thanks for pointing out the notation and reference issues,  and we will correct them as suggested in the revised manuscript.
> >
> > ## 5. Limitations
> >
> > ### Heterogeneous networks
> > Extension to heterogeneous networks is interesting. For example, a possible way is to perform RIM on each network view (defined by a single relation type) and maximize the aggregated influence across views.
> >
> > ### $\\small \\alpha$ estimation
> > In practice, we can estimate $\\small \\alpha$ with redundant votes across labelers (e.g., such as Amazon’s Mechanical Turk) by treating the majority vote as correct labels, like the Dawid-Skene algorithm [Dawid et al. 1979].
> >
> > [Dawid et al. 1979] A. P. Dawid and A. M. Skene. [Maximum likelihood estimation of observer error-rates usingthe em algorithm.](https://rss.onlinelibrary.wiley.com/doi/abs/10.2307/2346806) Journal of the Royal Statistical Society, Series C, pages 20–28, 1979.
> >
> > ### Minimizing risk
> > A good point. We actually connect the Eq.(1) with the RIM objective because maximizing the number of activated nodes actually reduces the generalization risk (as demonstrated in Figure 1). The reasonable condition for that is the common assumption used by both GCN and LP: connected nodes are likely to have the same label.

---

> > > ### Comment · Reviewer_qTkm · 2021-08-27
> > > **Response to clarifications**
> > >
> > > Thanks for the clarifications and additional experiments. I have raised my rating.

---

### Decision · Program_Chairs · 2021-09-27

**Decision:**

Accept (Spotlight)

**Comment:**

After some concerns of the reviewers could be resolved by the author feedback, the reviewers unanimously suggest to accept the submission. They generally found the submission interesting, theoretically sound, and well-presented. The authors should carefully incorporate the reviewer comments including their clarifications into the final version.